# Improving timeliness in the neglected tropical diseases preventive chemotherapy donation supply chain through information sharing: A retrospective empirical analysis

**Elena Kasparis**[1]*, **Yufei Huang**[2], **William Lin**[1], **Christos Vasilakis**[3]

**1** Department for Health, University of Bath, Bath, United Kingdom, **2** Trinity Business School, Trinity College Dublin, Dublin, Ireland, **3** School of Management, University of Bath, Bath, United Kingdom

* elena.kasparis@bath.edu

**Data Availability Statement:** The data underlying the results presented in the study are available from https://www.ntdeliver.com. While the system

## Abstract

### Background

Billions of doses of medicines are donated for mass drug administrations in support of the World Health Organization's "Roadmap to Implementation," which aims to control, eliminate, and eradicate Neglected Tropical Diseases (NTDs). The supply chain to deliver these medicines is complex, with fragmented data systems and limited visibility on performance. This study empirically evaluates the impact of an online supply chain performance measurement system, "NTDeliver," providing understanding of the value of information sharing towards the success of global health programs.

### Methods

Retrospective secondary data were extracted from NTDeliver, which included 1,484 shipments for four critical medicines ordered by over 100 countries between February 28, 2006 and December 31, 2018. We applied statistical regression models to analyze the impact on key performance metrics, comparing data before and after the system was implemented.

### Findings

The results suggest information sharing has a positive association with improvement for two key performance indicators: purchase order timeliness ($\beta = 0.941$, $p = 0.003$) and—most importantly—delivery timeliness ($\beta = 0.828$, $p = 0.027$). There is a positive association with improvement for three variables when the data are publicly shared: shipment timeliness ($\beta = 2.57$, $p = 0.001$), arrival timeliness ($\beta = 2.88$, $p = 0.003$), and delivery timeliness ($\beta = 2.82$, $p = 0.011$).

### Conclusions

Our findings suggest that information sharing between the NTD program partners via the NTDeliver system has a positive association with supply chain performance improvements, especially when data are shared publicly. Given the large volume of medicine and the

has a component requiring a login, the majority of the data can be found via the "Public Dashboards" options on the NTDeliver link and this data does not require a log-in. From the "Public Dashboards" option, "Search all orders," should be selected which will then route to this link: https://www.ntdeliver.com/report/search-all-orders?locale=en From there, the shipment data can be downloaded for the years and regions indicated in the study. Note that this data will not include the purchase order (PO) date or go signal dates, these two additional data points can be found by cross-referencing the purchase order number to the data found in the publicly-accessible country pages, which are accessed via this link: https://www.ntdeliver.com/country/

**Funding:** The authors received no specific funding for this work.

**Competing interests:** The authors have declared that no competing interests exist.

significant number of people requiring these medicines, information sharing has the potential to provide improvements to global health programs affecting the health of tens to hundreds of millions of people.

## Author summary

The supply chain to deliver donated preventive chemotherapy medicines is complex due to the many stakeholders and partnerships participating, as well as challenging because the logistics are further complicated by delivery to remote destinations in developing countries. As mass drug administration (MDA) campaigns involve treating hundreds of thousands to millions of patients in endemic regions within entire countries over the course of days or weeks, close coordination and timing of medicine delivery is critical. Inefficiencies caused by fragmented data systems and limited transparency on supply chain performance further challenges the ability to identify shipment issues and explore the root cause of the issues. Prior to 2016, delivery was performing below standards, lagging as much as 40% below the WHO target of 80% on-time delivery. These delays result in wasted medicine donations, increased program costs, delayed MDAs, or sometimes even completely missed MDAs. In September 2016, an online supply chain performance measurement system, "NTDeliver," was launched by the NTD Supply Chain Forum (a public-private partnership focused on managing and improving the preventive chemotherapy (PC) donation supply chain) to enhance supply chain performance and information transparency. Our findings suggest that information sharing through NTDeliver is positively associated with performance improvements at key stages in NTD supply chain, especially when information is made publicly accessible, focused on access for country program managers. The study findings support investment in supply chain systems and commitment to data transparency, in the context of a growing focus on supply chain investment in NTD programs.

## Introduction

Public-private partnership programs provide medicines for preventive chemotherapy (PC) through mass drug administration (MDA) campaigns to more than one billion people annually. The programs are sustained by large-scale donations from major pharmaceutical companies in support of the World Health Organization's 2012 "Roadmap to Implementation," which outlined global strategies and 2020 targets to control, eliminate, and eradicate Neglected Tropical Diseases (NTDs) [1]. While significant progress was made towards these 2020 targets, the WHO has recently released a new NTD roadmap with 2030 targets, which many pharmaceutical manufacturers have committed to continuing to support [2]. As of January 2020, 15 billion doses of medicines were donated towards these PC-NTD programs [3]. The donations from pharmaceutical companies are what makes these the world's largest and most successful public health programs [4]. MDA campaigns are comprised of once or twice-a-year treatment with one or more medicines at the community level that bring together a number of stakeholders, requiring considerable coordination as they typically involve treating hundreds of thousands to millions of patients in endemic regions within entire countries over the course of days or weeks [5].

The logistics involved to make these medicines available to support MDAs is both critical and complex, due to the many stakeholders and partnerships involved in meeting the targeted treatment date. Fig 1 provides an overview of the processes involved and performance measures associated with each link in the supply chain. With the considerable resources and coordination involved within a narrow timeframe, inefficiencies caused by fragmented data systems and a lack of visibility to supply chain performance have resulted in substandard performance for on-time delivery of medicine to in-country central medical stores. Sometimes delivery targets may lag as much as 40% below the WHO target for 80% of all shipments to be delivered at least one month before the planned MDA date [6]. Delivery delays result in waste, increased program costs, and delayed or even completely missed MDAs, leaving communities susceptible to infections and disease recrudescence [7].

To improve the efficiency and performance of NTD supply chain, the NTD Supply Chain Forum was established in 2012 [3]. The NTD Supply Chain Forum is comprised of NTD supply chain experts from the WHO, pharmaceutical companies, nongovernmental organizations,

**Application and Order**
- **Process:** application/order finalized at country/regional level (at least 8 months prior to Mass Drug Administration - MDA); WHO HQ raises purchase order to the manufacturer (at least 6 months prior to MDA)
- **Performance measure:** timely submission of application against dates; timely purchase order issue (at least six months prior to Mass Drug Administration – MDA)
- **Involved partners:** Ministry of Health/NTD coordinator, WHO, pharmaceutical manufacturers

**Manufacturing**
- **Process:** pharmaceutical manufacturers produce medications and package existing stock for shipment
- **Performance measure:** packing timeliness - no specific standard; performance measures are variable by manufacturer
- **Involved partners:** WHO, pharmaceutical manufacturers, third party logistics service providers, supporting non-governmental organizations

**Shipment**
- **Process:** "go signal" obtained from WHO to initiate shipment; shipped by air, sea, or land
- **Performance measure(s):** go signal timeliness (actual date against request date); timely shipment (at least three months prior to MDA)
- **Involved partners:** WHO, pharmaceutical manufacturers, third party logistics service providers, supporting non-governmental organizations

**In-country Processes: Arrival to Distribution via MDA**
- **Process:** shipment arrives in country, clears customs, delivered to central medical store; medicines will be distributed to district hospitals, health outposts, or program campaigns to be distributed via MDAs
- **Performance measure(s):** arrival timeliness - no specific standard; delivery to central medical store at least one month in advance of MDA; other last mile performance measures may vary by country
- **Involved partners:** WHO, pharmaceutical manufacturers, third party logistics service providers, customs agents, supporting non-governmental organizations, Ministry of Health/NTD coordinator,

**Fig 1. Components of the NTD supply chain from application for donation to in-country delivery to central medical stores.**

donor organizations, ministries of health, and logistics providers [3]. Subsequently to the formation of this forum, "NTDeliver," a centralized information system, was launched in 2016 by the NTD Supply Chain Forum to share data from various partners along supply process chain as means of facilitating performance information sharing [8]. Through NTDeliver, all stakeholders in the supply chain—and even the general public—may access performance metrics on all shipments of the four medicines that participate in data sharing through NTDeliver.

Information sharing in the context of supply chains—"the extent to which crucial and/or proprietary information is available to members of the supply chain"—is an integral aspect of performance management and the sharing of accurate and timely information has been linked to supply chain performance improvements [9,10]. Many studies have been conducted on the value of information sharing to improve supply chain performance in the private sector. Information sharing has been proven to have a range of benefits, from improved resource utilization to reduced cycle time between order and delivery [11]. These benefits stem from the increased transparency that enable risks to be anticipated and shared among supply chain partners, which strengthens coordination to achieve optimal operational performance [11–13]. Largely, this body of literature with empirical studies investigates the value of information sharing in a commercial supply chain focused on a dyadic relationship, primarily between two partners, a buyer and a supplier [14].

Despite the growing importance of supply chain initiatives in the global NTD agenda, there is limited research dedicated to exploring measures to improve NTD supply chain performance nor the impact of information sharing. This limitation may be an indicator that performance measurement and management systems have not been widely developed and systematically implemented as part of the overall humanitarian supply chain strategy [15]. Only Korpoc et al's 2015 research on the impact of the NTD "first mile" processes (the segment of the NTD supply chain covering up to delivery to central medical stores) on MDA timeliness acknowledges this area of NTD supply chain performance measurement by identifying the need for performance indicators and outlining suggested metrics [7]. Furthermore, the recent COVID-19 pandemic has raised the profile both of the criticality of publicly sharing timely data in the global health domain and the topic of assuring robust supply chains to meet global health goals [16–21]. Thus, the timing could not be better to study information sharing and supply chain management in the wider global humanitarian health context.

This study seeks to evaluate empirically the impact of information sharing via NTDeliver on supply chain performance—improvements which ultimately contribute to achieving the global NTD targets. We examine the following two research questions: 1) what is the impact of information sharing through NTDeliver on the performance of the NTD PC medicines donation supply chain? and 2) what is the impact on the performance when country-level data are made publicly accessible? We use data obtained from the NTD Supply Chain Forum and implement regression models on these research questions. We find that information sharing has a positive association with improvement to two performance indicators of the NTD supply chain: purchase order timeliness and delivery timeliness. Furthermore, when country-level information sharing is made publicly accessible, a positive association is observed primarily on three downstream indicators: shipment timeliness, arrival timeliness, and delivery timeliness.

## Methods

### Study design and scope

We used retrospective data from NTDeliver that are routinely collected from and managed by supply chain partners supporting delivery of PC medicines from pharmaceutical manufacturing facilities to central medical stores. Permission was granted to use this data by the NTD

Supply Chain Forum. The data are derived from vetted, existing data sources managed by NTD supply chain partners, such as:

- WHO Preventive Chemotherapy and Transmission databank

- Data provided to WHO country offices by the countries' Ministry of Health through the joint application process to request donations

- Purchase orders raised by the WHO headquarters

- Shipping documents generated by logistics service providers, in partnership with pharmaceutical donors

The data represent shipments of four medicines from four manufacturers to treat three different diseases, accounting for almost 11.5 billion doses of PC medicines to 103 recipient countries covering 1,484 total shipments from February 28, 2006, to December 31, 2018. The data are refreshed and uploaded from these various sources daily [22].

While there are numerous medicines for NTDs donated by numerous pharmaceutical manufacturers for the NTDs, this research's scope focuses on PC medicines donations managed by the WHO through the "joint application package" (JAP) established in 2013, which supports an integrated review and subsequent reporting on medicines usage [23]. The JAP streamlines the application for donation of multiple medicines, especially as medicines are co-administered where diseases are co-endemic [23]. These PC medicines include: diethylcarbamazine citrate, albendazole, mebendazole, and praziquantel [24]. This focus is justified by the considerable volume of medicines, the unique nature of this supply chain that includes WHO involvement, the importance of PC to achieve NTD targets, and accessibility of data through NTDeliver. There are opportunities to improve processes across the supply chain, but the focus of this research will be on the segment of the supply chain that entails delivery from pharmaceutical manufacturing facilities to central medical stores, also referred to by partners as the "first mile" [7]. We chose to focus on the first mile due to the WHO drive to improve on-time delivery to central medical stores, the accessibility of relevant data, and opportunity to leverage information sharing among the many partners involved in this segment. Central medical stores (CMS) (most commonly utilized in Africa, Asia, and Latin America) serve as a warehouse and administrative facility that receives, stores, and manages medical supplies for national health programs and initiatives and are generally leveraged for humanitarian stock [6,25]. Improving the on-time delivery of PC medicine to the CMS is critical to the downstream in-country distribution to program sites where they are needed [6]. While the last mile (which encompasses drugs transport from central medical stores to district-level stores then on to MDA distribution) certainly involves many challenges that contribute to MDA delays, issues with the first mile have shown in practice and research to have a downstream effect on the last mile and ultimately on MDA timeliness [3,7]. Thus, alleviating first mile issues will be a step in the right direction to improve MDA timeliness.

## Variables

The variables for this analysis are various key performance indicators (KPIs), which are actively reviewed through the NTD Supply Chain Forum and of interest to the supply chain partners, including the WHO. The most critical KPI is delivery timeliness, by which the WHO evaluates performance of this first mile of the NTD supply chain [6]. Both the independent and dependent variables were created from the data in the system, reflecting the KPIs and benchmarks standards tracked by the NTD Supply Chain Forum partners. Table 1 summarizes the key variables in the analysis.

**Table 1. Variable definitions.**

| Variable type | Label | Characteristics/description |
|---|---|---|
| Independent | NTDeliver | *Binary (or dichotomous) variable* distinguishing whether the donation order has been tracked in NTDeliver |
| | NTDeliver public access | *Binary (or dichotomous) variable* distinguishing whether country-level data was publicly accessible in NTDeliver |
| Dependent | PO timeliness | *Continuous variable* measuring the timeliness of the Purchase Order (PO) provided by the WHO HQ to initiate the order for the pharmaceutical donation; measured in months |
| | Go signal timeliness | *Continuous variable* approval from the WHO for the order to be shipped, once the needed documentation is prepared; calculated by finding the number of days between go signal request date and go signal approval date; measured in days |
| | Shipment timeliness | *Continuous variable* measuring the timeliness of the order being shipped from the manufacturing site; calculated by finding the number of months between the shipment date and the MDA; measured in months |
| | Arrival timeliness | *Continuous variable* measuring the timeliness of the arrival of the shipment into the country; calculated by finding the number of months between the arrival date and the MDA; measured in months |
| | Delivery timeliness | *Continuous variable* measuring the timeliness of the shipment's arrival to the delivery warehouse; calculated by finding the amount of time between the delivery date and the MDA; measured in months |
| Control | WHO region | *Categorical variable* which identifies the WHO region the ordering country is associated with; regions include Regional Office for Africa (AFRO), Regional Office for the Americas (AMRO), Regional Office for the Eastern Mediterranean (EMRO), Regional Office for Europe (EURO), Regional Office for Southeast Asia (SEARO), Regional Office for the Western Pacific (WPRO) |
| | International Logistics Performance Index (LPI) | *Continuous variable* which provides the score from the benchmark tool that the World Bank manages to measure performance of the logistics supply chain within a country, providing a qualitative evaluation by trading partners of the country's logistics performance |
| | Medicine type | *Categorical variable* which identifies the specific donated medicine ordered; four medicines included: albendazole (ALB), diethylcarbamazine citrate (DEC), mebendazole (MEB) and praziquantel (PZQ); each specific medicine is associated with one pharmaceutical manufacturer, but may be used to treat more than one disease |
| | Disease treated | *Categorical variable* which identifies the disease(s) the medicine is used to treat; three diseases included: lymphatic filariasis (LF), schistosomiasis (SCH), or soil-transmitted helminthiases (STH) |
| | Order size | *Categorical variable* which identifies number of tablets ordered; segmented into three groups: ≥10M; <10M and ≥1M; <1M |
| | Mode of shipment | *Categorical variable* which identifies the mode of shipment for the given order; one of three modes identified for each order: air, sea, or land |

Important cofactors, incorporated via control variables, were included in the analyses to explore whether the relationship of the independent and dependent variables is skewed or invalidated by other factors. The WHO region was also included as a control since performance may vary according to the destination (shipment routes and customs clearance processes vary according to the destination) and to account for any regional-level improvement initiatives that could also explain performance improvements. Controlling for region may be considered more meaningful than controlling for country since regional-level improvements are more likely to impact the results as regional efforts likely impact all countries under their purview and therefore a great scope of the data sample. Yet, there is value to control for country logistics factors. While the NTD Supply Chain Forum has limited visibility of the country-level logistics since the focus is on the first mile, we can incorporate an external assessment of country logistics factors as a control variable. We believe the World Bank international Logistics Performance Index (LPI) benchmark tool is a relevant indicator to help control for such factors. This score measures performance of the logistics supply chain within a country and provides a qualitative evaluation by trading partners of the country's logistics performance [26].

The controls for medicine and disease were included with consideration for the fact that the medicines are produced by different manufacturers for different disease programs, which may lend itself to some variability in the supply chains. Mode of shipment was also

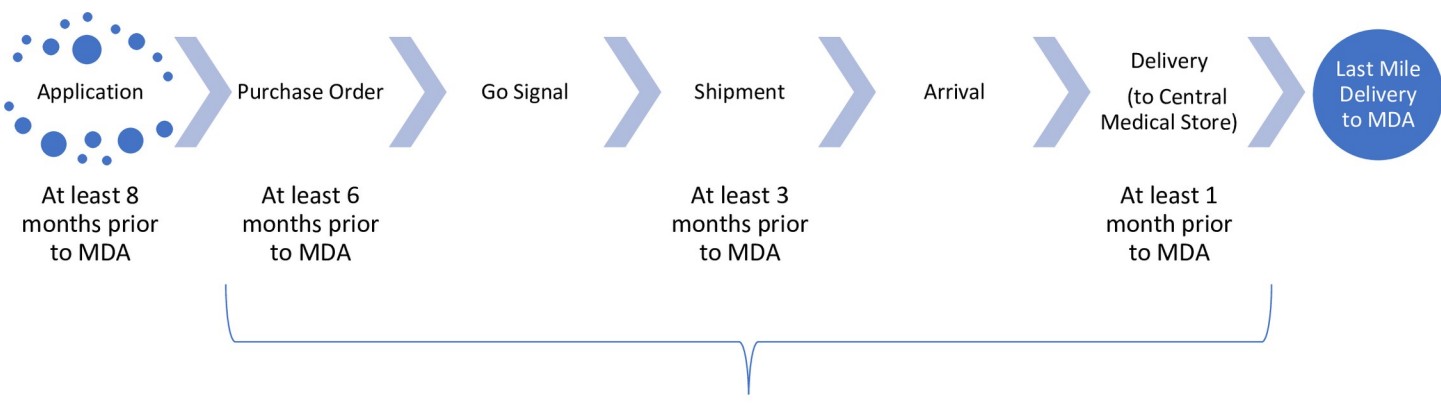

**Fig 2. Simplified timeline of the NTD supply chain, associated benchmarks, and scope of research.**

incorporated as a control since the shipment speed varies between air, land, and sea. Lastly, order size was also included even though the supply chain process is the same regardless of order size; larger orders could take more time to prepare and smaller orders are often shipped by air, making this also a necessary factor to control. Details on the coding of the control variables can be found in S1 and S2 Tables.

Connecting back to the full process shared in Fig 1, Fig 2 provides a diagram illustrating the process flow in a summary view and the components in/out of scope of the study, along with reference to any targeted timeline benchmarks as relevant.

## Statistical analysis

A quasi-experiment design, using a "one-group pretest-posttest design without control group," was chosen as the research was initiated a few years after the launch of NTDeliver. In addition, NTDeliver was implemented in a real-world application that did not roll out the system in a phased approach, barring any ability to conduct randomization. This design was used to leverage historical data available on performance to understand the impact of this intervention. An ordinary least squares (OLS) regression model was used to review the relationship between implementing NTDeliver and its impact on delivery timeliness and other KPIs. Furthermore, review of the data confirms that the normality of the error distribution assumption for OLS regression is met. Q-Q plots were used to verify that most data are on a distribution lying on approximately on a straight line. While go signal timeliness did not show a straight trend, the central limit theorem enables the normality assumption to be met in the case of a "sufficiently large sample," for which the literature generally notes a sample of $>50$ would be "robust to violation of the normality assumption" [27]. This variable had over 200 samples, thus meeting the central limit theorem condition. While both medicine and disease type are included as control variables, these variables are not found to be significantly correlated since some medicines treat more than one disease and therefore is not a 1:1 mapping; hence, multi collinearity is not a concern. Control variables and robustness checks were applied to strengthen the validity of the results. We considered p values less than 0.05 to be statistically significant.

The shipment data extracted from the system cover orders made through December 31, 2018 and was therefore segmented in two groups to address the first research question: 1) shipments with POs raised prior to the implementation of NTDeliver (February 28, 2006-August

31, 2016) for the "pre" NTDeliver group; 2) shipments with POs raised after the implementation of NTDeliver (September 1, 2016-Dec 31, 2018) for the "post" NTDeliver group. Only the data in the "post" group were used to answer the second research question regarding the impact on shipment performance of making country-level data publicly accessible. The data in the "post" group were split into two groups, with consideration to February 1, 2018, as the implementation date of this publicly accessible data: 1) "Post 1" = 0 for shipments with a PO date prior to February 1, 2018 but after August 31, 2016; 2) "Post 2" = 1 for shipments with a PO date equal to or later than February 1, 2018 but earlier than January 1, 2019.

## Results

We first study the impact of information sharing through NTDeliver on shipment performance within the NTD PC medicines supply chain. The data collected included 1,484 total shipments, with 1,068 shipments classified in the "pre" NTDeliver group and 416 in the "post" group. As noted, pairwise deletion was used in cases where data were missing, which accounts for the differing number of observations between variables. Table 2 summarizes the regression results illustrating the bivariate associations between the implementation of the NTDeliver system and NTD supply chain KPIs. Complete regression results are included in Table A3 in the S3 Table.

Notably, delivery timeliness, the KPI considered most important to measure supply chain performance, shows a positive and significant association with a *p*-value of 0.027. PO timeliness also demonstrates positive and significant results supporting the hypothesized direction

Conversely, the results for go signal timeliness appear to support the null hypothesis. While go signal timeliness is highly significant (p<0.001), even though the coefficient value is positive, this actually indicates a negative relationship with information sharing due to the calculation method for the variable (calculated as the difference between requested and actual go signal date). The coefficient suggests an increase in the difference between the go signal request and approval of about 40 days. Because there is no perceivable standard for establishing this request date and it is defined per request of the pharmaceutical manufacturer and any supporting partners, additional analysis provides further insight into how the go signal timeliness calculation may have changed after NTDeliver was implemented.

These results provided indicate that the difference between PO date and go signal request date has significantly decreased, since the coefficient is negative and therefore indicate changes to how the request date was defined (Table 3).

Next, we study the impact of making country-level data publicly accessible and particularly promoting this data access to country program managers, as training sessions were also provided for country program managers to publicize the release of this data and provide education

**Table 2. Regression results showing the relationship between NTD supply chain key performance indicators and implementation of information sharing through NTDeliver.**

| Variables | N | B reg coefficient | p-value | Predicted direction | Actual direction |
|---|---|---|---|---|---|
| PO timeliness (in months) | 897 | 0.941 | 0.003 | ↑ | ↑ |
| Go signal timeliness (in days) | 216 | 41.483 | < .001 | ↑ | ↓ |
| Shipment timeliness (in months) | 535 | 0.033 | 0.927 | ↑ | ↑ NS |
| Arrival timeliness (in months) | 661 | 0.516 | 0.129 | ↑ | ↑ NS |
| Delivery timeliness (in months) | 528 | 0.828 | 0.027 | ↑ | ↑ |

Abbreviations: Reg, regression; NS, non-significant

Incorporated control variables: WHO region, international LPI, medicine type, disease, order size, mode of shipment

**Table 3. Bivariate association between regression results showing the relationship between implementation of information sharing through NTDeliver and key differences between go signal dates.**

| Variables | N | B reg coefficient | p-value |
|---|---|---|---|
| Difference between PO date and go signal request date (in days) | 263 | -48.862 | <0.001 |
| Difference between go signal approved date and shipment date (in days) | 263 | -50.899 | <0.001 |

Abbreviations: Reg, regression; NS, non-significant

on effective usage. The hypothesis is that extending information sharing may have a positive impact on the downstream processes that are actively displayed in the public country pages, which provides country managers with shipment, arrival, and delivery statuses. Table 4 provides the analysis results to answer the second research question regarding the impact of making country-level data publicly accessible.

The results show three variables with significant, positive association: shipment timeliness, arrival timeliness, and delivery timeliness—consistent with the hypothesis that these variables would be impacted since they are actively displayed in these publicly accessible pages. In the main regression results, shipment timeliness and arrival timeliness did not show any significance from the information sharing. In these results, shipment timeliness is significant at a *p*-value of 0.001 and with a substantial coefficient of 2.57. Arrival timeliness is significant with a *p*-value of 0.003 and a coefficient of 2.88. Lastly, the important delivery timeliness performance indicator is also significant with a *p*-value of 0.011 and a coefficient of 2.82.

We also conducted further analyses to check the robustness of the results (see S4 Table). First, we re-ran the regression accounting for a lag in impact from information sharing. The analysis revealed that that all dependent variables, except arrival timeliness, remain significant and generally consistent with the main results when accounting for a six-month theoretical "lag time to benefit," considered as the time between implementing the intervention and observing improved outcomes [28]. Additionally, we conducted a "double pretest" to test the validity of the "pre" group as comparison. This "double pretest" was used as a validity check to ensure that merely "history" and/or "maturation" is not the reason for the differences between the pretest and posttest results, rather than independent variable [29]. The pretest group was divided and compared for significant differences in performance to assure any differences in performance within the pretest group are minimal and/or less than the difference between the pretest and posttest groups. The groups were divided with roughly the same number of shipments in each group, with one group comprised of shipments with POs raised between 2006–2013 and the other with POs raised between 2014–2016 August. The results from this analysis indicated that only PO timeliness had a positive, significant difference between these two pretest groups. No other dependent variables have an observable, significant difference in performance.

**Table 4. Regression results showing the relationship between implementation of publicly accessible country-level data through NTDeliver and NTD supply chain key performance indicators.**

| Variables | N | B reg coefficient | p-value | Predicted direction | Actual direction |
|---|---|---|---|---|---|
| PO timeliness (in months) | 387 | 0.33 | 0.49 | NS | ↑NS |
| Go signal timeliness (in days) | 132 | -14.88 | 0.42 | NS | ↑NS |
| Shipment timeliness (in months) | 296 | 2.57 | 0.001 | ↑ | ↑ |
| Arrival timeliness (in months) | 262 | 2.88 | 0.003 | ↑ | ↑ |
| Delivery timeliness (in months) | 255 | 2.82 | 0.011 | ↑ | ↑ |

Abbreviations: Reg, regression; NS, non-significant

## Discussion

Lack of coordination and limited transparency are two top issues in humanitarian supply chains [30]. Although existing literature on for-profit supply chains suggests addressing the issues though information sharing, it is unclear whether and how information sharing can improve performance in the non-profit humanitarian context [10–13]. In this paper, we examined the potential impact of information sharing on humanitarian supply chain performance. Our analysis is the first to undertake an empirical study evaluating performance and information sharing both in the context of the NTD supply chain and in the broader humanitarian space. While most existing literature focuses on information sharing in for-profit supply chains, which is typically focused on sharing information between the buyer and supplier, this paper contributes to the existing literature and addresses the gap by investigating the impact of sharing information publicly for non-profit supply chains. The results of our study demonstrate the value of investment in supply chain performance measurement and information sharing towards the success of global health partnerships and such initiatives may be implemented in the broader context of humanitarian programs.

We find that information sharing is positively associated with the timeliness of several key stages in NTD supply chain, i.e., PO timeliness, arrival timeliness, shipment timeliness, and the key success measure for the NTD supply chain—delivery timeliness. Delivery timeliness, arguably the most critical KPI to measure first mile performance, appears to have a positive and significant association with information sharing. The analysis showed that in the post implementation phase, on average, completed POs were submitted one month earlier. In addition, go-signals were issued expeditiously after POs were issued. Improvements in these two metrics possibly contributed to the observed results of earlier delivery of medicines to the CMS, suggesting that information sharing as a result of the NTDeliver may be a factor.

Furthermore, information sharing appears to be possibly more impactful when information is released publicly and particularly promoted to country program managers, compared to when it is shared only with the supply chain partners (i.e., WHO and pharmaceutical donors). A significant positive association was evident for three variables upon this information extension: shipment, arrival, and delivery timeliness Neither shipment nor arrival timeliness met the threshold for significance in the first set of results, but both variables meet this threshold in these additional results. Delivery timeliness continues to meet the threshold for significance in these results, but also with a coefficient substantially larger than in the first set of results. While there is clearly an impact from the implementation of NTDeliver without the country pages, the addition of the country pages appeared to extend the impact to downstream processes after PO creation and the go signal. As previously noted, this result was hypothesized since the country pages specifically provide access to status from the point of shipment. Thus, it is logical that we did not see in the results significant improvements to upstream KPIs such as PO timeliness and go signal timeliness, since they are not featured in these country pages. These results suggest that it may be more effective to extend information sharing to stakeholders responsible for performance across the entire supply chain.

The robustness checks also showed that even if the information sharing effect took time to make an impact, the performance still improved and can be associated more confidently with the implementation of NTDeliver. Furthermore, although there is no "control group" in our research design, we conduct a "double pretest" to investigate if other observed time-varying variables may contribute to the significant results. Comparison with performance before the information sharing was implemented suggests that the supply chain performance does not simply improve over time, with exception to one performance indicator. Only PO timeliness indicates a significant positive change over time during the period before NTDeliver was

implemented. The reason for this change in PO timeliness may be attributed to a process change coinciding around the timing of the second pretest group defined in this analysis, which included medicines ordered from 2014 onward: the JAP was implemented by the WHO in 2013 to standardize processes to support an integrated application submission and review for donated medicines [23]. This JAP process most positively impacts the PO process as it promoted more coordination between the various levels of WHO offices to assure timely applications and order fulfillment for donated medicines [23].

This research has some limitations that may naturally inform future research. The quasi-experimental design used lacks a control group and random assignment since NTDeliver was implemented for all PC medicine donations managed through the NTD Supply Chain Forum [31]. Although we used a "double pretest" robustness check to verify our results, future research could study the impact of information sharing in a controlled setting with randomization incorporated in the design. Also, modifications to the design could entail incorporating qualitative research using interviews and/or surveys to investigate whether stakeholder behavior changes may be drivers for observed performances changes. Furthermore, there is a growing desire for financial donors to understand the impact of investments from an outcomes perspective, especially in the interest of funding effective health innovations that offer value for money [32].While our research results certainly helps to validate the positive association between information sharing and supply chain performance, further research on how the supply chain performance improvements in this first mile result in a reduction in delayed and/or missed MDAs would provide more perspectives on linking the delivery timeliness improvements to the number of additional individuals reached. Also, incorporating any data pertaining to the last mile aspect of the NTD supply chain in NTDeliver might help to improve those downstream processes, which remains a priority, given the last mile's impact on efficient and effective distribution and planning for MDAs [33]. Lastly, with respect to the current global health climate, the COVID-19 pandemic had a significant impact on NTD programs, with the WHO recommending postponing MDAs to respect public health measures that advocate for physical distancing to slow the spread of the virus [32]. Further research is needed to gain insight on new challenges from these disruptions to understand the impact by region and how information sharing may help to mitigate such disruption and support managing uncertainty for global health campaign supply chain planning during a pandemic.

Our results have practical implications for NTD supply chain management practices. As the deadline approaches for achieving the 2030 targets set out in the new WHO roadmap and the relevant NTD goals in target 3.3 of the Sustainable Development Goals, there is a high degree of confidence that these results affirm that investment in supply chain information sharing is a critical to ensuring success. In fact, the new WHO roadmap dedicates an entire section to "Access and logistics," in which supply chain management priorities for improvement are outlined under the umbrella concept that "effective supply chain management is vital to ensuring access to quality-assured NTD medicines and other products" [34]. Given the relationship between first mile supply chain performance and timeliness of MDAs, investment in supply chain information sharing is worthwhile to support improvements to NTD program management [7].

Furthermore, the findings imply additional benefits of information sharing when extending information sharing to a broader audience, particularly focusing on program managers. Incorporating visibility to upstream data, such as attaching country applications or tracking the regional office approval date, may improve these processes as well. Such upstream processes have been noted as potentially impacting delivery timeliness by the WHO HQ—such as the fairly significant time taken for the regions to review applications—and further incorporation of these processes in NTDeliver may benefit the end-to-end NTD supply chain [7].

Given the significant volume of medicines and the number of people requiring these medicines, the research implications have the potential to impact global health programs affecting the health of tens to hundreds of millions of people. The research supports that, even in absence of financial remuneration, information sharing contributes measurable supply chain improvements and supports investing further in performance measurement in humanitarian supply chains. Beyond the NTD space, data transparency is generally viewed as a challenge with country governments citing national sovereignty and privacy in refusing to release data for public consumption [35]. Positive results from extending the information sharing argue in favor of the value and benefits of information sharing in the global health space. With the profile and importance of the supply chain continues to elevate in humanitarian programs, especially those in the healthcare space, there will be an opportunity to invest further in such performance measurement tools to bring more evidence-based approaches to decision making. This has significant potential to promote accountability and coordination resulting in goals achieved and improved health outcomes. As the global health aid landscape is becoming more focused on driving measurable performance and impact from investments, these findings support investing in supply chain systems and commitment to data transparency.

## Supporting information

**S1 Table. Dummy coding matrices.**
(DOCX)

**S2 Table. International Logistics Performance Index (LPI) Control Variable Assignment Details.**
(DOCX)

**S3 Table. Full linear regression results.**
(DOCX)

**S4 Table. Robustness check results.**
(DOCX)

## Acknowledgments

The authors would like to acknowledge the NTD Supply Chain Forum for generously providing access to the data used in this research via NTDeliver. In particular, a big thank you is extended to key leaders of this Forum, including Tijana Williams, Dr. Christian Schröter, and Cassandra Holloway for support on granting this data access and disseminating results to the NTD supply chain partners of the Forum. We also thank TJ Muehleman and Emily Tunggala from the NTD Supply Chain Forum technology partner, Standard Co, for support to confirm details on the data sources in NTDeliver and additional insights on the data storage and management.

## Author Contributions

**Conceptualization:** Elena Kasparis, Yufei Huang.

**Data curation:** Elena Kasparis.

**Formal analysis:** Elena Kasparis, Yufei Huang.

**Investigation:** Elena Kasparis.

**Methodology:** Elena Kasparis, Yufei Huang, William Lin, Christos Vasilakis.

**Project administration:** Elena Kasparis.

**Resources:** Elena Kasparis.

**Software:** Elena Kasparis.

**Supervision:** Yufei Huang, William Lin, Christos Vasilakis.

**Validation:** Elena Kasparis, Yufei Huang.

**Visualization:** Elena Kasparis.

**Writing – original draft:** Elena Kasparis.

**Writing – review & editing:** Elena Kasparis, Yufei Huang, William Lin, Christos Vasilakis.

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
