## [Decision Letter · Decision Letter 0]

30 Aug 2021

Dear Dr. Kasparis,

Thank you very much for submitting your manuscript "Improving Timeliness in the Neglected Tropical Diseases Preventive Chemotherapy Donation Supply Chain through Information Sharing: A Retrospective Empirical Analysis" for consideration at PLOS Neglected Tropical Diseases. As with all papers reviewed by the journal, your manuscript was reviewed by members of the editorial board and by several independent reviewers. In light of the reviews (below this email), we would like to invite the resubmission of a significantly-revised version that takes into account the reviewers' comments. 

We cannot make any decision about publication until we have seen the revised manuscript and your response to the reviewers' comments. Your revised manuscript is also likely to be sent to reviewers for further evaluation.

Sincerely,

Gregory Deye

Associate Editor

Bruce Lee

Deputy Editor

Reviewer's Responses to Questions

**Key Review Criteria Required for Acceptance?**

**Methods**

-Are the objectives of the study clearly articulated with a clear testable hypothesis stated?

-Is the study design appropriate to address the stated objectives?

-Is the population clearly described and appropriate for the hypothesis being tested?

-Is the sample size sufficient to ensure adequate power to address the hypothesis being tested?

-Were correct statistical analysis used to support conclusions?

-Are there concerns about ethical or regulatory requirements being met?

Reviewer #1: see below

Reviewer #2: The study objectives are clear and there is a clear set of testable hypotheses.

However, the study design does not take into account several factors:

-Were there any time-varying factors besides the NTDDeliver and NTDPublic which may be coincident with the study time frame. Countries, manufacturers, and WHO are constantly trying new improvements. Not clear how the study design robustly controls for that?

-Presumably there is a higher likelihood of missing data in the pre-intervention sample. So without explicitly showing that missing data would be completely random (and covariance matrix etc.) it is unclear how the study design removes biases that may result from non random missing data.

-An important control variable which is missing is country characteristics e.g. country logistics factors, etc. What if there was a differences in the countries receiving the orders in the pre-and post intervention phase. The authors have controlled for WHO region, but that is too coarse a variable to account for country specific differences

-Did the authors explore the use of a RD type study design?

Reviewer #3: No issues

**Results**

-Does the analysis presented match the analysis plan?

-Are the results clearly and completely presented?

-Are the figures (Tables, Images) of sufficient quality for clarity?

Reviewer #1: see below

Reviewer #2: The analysis matches what the study plan was.

The result are difficult to interpret without a clear exposition of what each of the time variables means ie. when does the clock start for each timeliness variables and then does the clock stop. The authors should consider creating a diagram with the timelines and variable definition on that.

Reviewer #3: No issues

**Conclusions**

-Are the conclusions supported by the data presented?

-Are the limitations of analysis clearly described?

-Do the authors discuss how these data can be helpful to advance our understanding of the topic under study?

-Is public health relevance addressed?

Reviewer #1: see below

Reviewer #2: the conclusions are supported by the data with the caveats in the study design outlined earlier.

The work has significant relevance for public health but needs to be more robust in its analysis

Reviewer #3: -Based on the findings can the authors further clarify which stakeholder behavior changed based on the variable. E.g., Did PO timeliness reflect WHO management? Did shipping times reflect the pharmaceutical companies. Were these results possibly influenced by other stakeholders indirectly such as NGOs supporting governments efforts?

**Editorial and Data Presentation Modifications?**

Reviewer #1: see below

Reviewer #2: (No Response)

Reviewer #3: CLARIFICATION IN THE INTRODUCTION

• “While significant progress was made towards these 2020 targets, the WHO has recently released a new NTD roadmap with 2030 targets, which pharmaceutical manufacturers have committed to continuing to support [2].” Pharmaceutical companies support the road map but not all donation programs have committed to donating pharmaceuticals through 2030. 

• “Delivery delays result in waste, increased program costs, and delayed or even completely missed MDAs, leaving individuals susceptible to NTD infections [7].” More accurate to state “…..leaving communities susceptible to disease recrudescence.” 

• “Figure 1. Components of the NTD supply chain from application for donation to in-country delivery to central medical stores”. 

- Application: performance measure is 8 months prior to MDA (not 6)

- Manufacturing: performance measures are variable by manufacturer

- Shipment: three months prior to MDA is the UNICEF standard of 10 years ago for sea freight. This is not WHO policy. 

- In-country processes: Performance measures should say varies by country

Clarify the NTD Supply Chain Forum membership. If stating the present Forum makeup, some members are missing. If wishing to indicate the makeup at the time of establishment, please make this clear.

**Summary and General Comments**

Reviewer #1: Overall, this is an interesting manuscript that attempts to utilize a pre-post design to evaluate the potential impact of the NTDeliver platform on improving supply chain management for PC diseases. The findings are of interest to the NTD funder, policy and program community and fit well within the scope of this journal. There are substantial issues with the way in which the paper is presented and written that need to be addressed. In addition, the authors need to tone down the language implying causality, as this study design can only assess associations, not causality. If these changes are made, I believe the paper will be a useful contribution to the literature.

A critical limitation of this analysis is that it only evaluates the impact on the “first mile”. It is well known that many of the delays in getting MDA drugs to communities happen between the central store and the program. As such, it is difficult to know how much of an impact this platform actually has on drug delivery. In addition, it does not seem logical to evaluate timeliness as a continuous measure – it probably makes no difference if the drugs are in the central warehouse 6 months or 5 months ahead of MDA – what matters is whether they are there and can be distributed in time for MDA. I would think a binary cutoff based on what this sufficient time is would be more practical. Finally, some of the “improvements” may be statistically significant but that does not make them programmatically relevant. I am not sure a two week improvement is meaningful for example.

The paper is written from the standpoint of a data analyst and much of the language is difficult to follow and not clear to the intended audience. The way in which the analysis is presented is not intuitive and does not align with metrics that the audience will want to understand. 

The authors seem to erroneously attribute magnitude of significance with the p value in their analysis. First, it is not at all clear why a non-standard p value of .1 was chosen to represent significance, particularly given that the study design is already weak. This seems likely to have been a post-hoc decision made following a look at the results, particularly given that many analyses are right at the 0.9 level. Please revise to use a standard p value of 0.5 for significance. In addition, something is not “more significant” if its p value is smaller. The key metric is the effect size, and then whether or not it meets the threshold for significance. Please revise the text accordingly.

The results and the discussion need to be separated. There is far too much discussion included in the results section. Please just present the results cleanly and then interpret them in the discussion.

The weaknesses of this study design are a major limitation (as noted by the authors). There is no reason why an RCT or a stepped wedge study could not have been designed in coordination with WHO. One key takeaway in the conclusion is the need to plan robust evaluations from the start so that more meaningful impact assessment can be conducted.

I was confused by the language around confounders and controls – these are important cofactors or mediators of effect – not confounders.

The data could be presented graphically or at least in some much more readily digestible format. The raw output of the regression analyses are not the best way to present these data to the intended audience.

Small issue – data are pleural – datum is singular – update text accordingly.

Reviewer #2: This paper addresses and important and very interesting area for policy relevance for NTDs.

The authors utilize an interesting data set and define performance measures with direct operational relevance to NTD programs and NTD supply chains

However, there are multiple deficiencies in the study design which need to be addressed. It is important for this study to be robust in its methods for the findings to be taken seriously for further investment in the interventions analyzed.

Reviewer #3: (No Response)

PLOS authors have the option to publish the peer review history of their article (what does this mean?). If published, this will include your full peer review and any attached files.

Reviewer #1: No

Reviewer #2: No

Reviewer #3: No
---

## [Editor Report · Decision Letter 1]

5 Nov 2021

Dear Dr. Kasparis,

We are pleased to inform you that your manuscript 'Improving Timeliness in the Neglected Tropical Diseases Preventive Chemotherapy Donation Supply Chain through Information Sharing: A Retrospective Empirical Analysis' has been provisionally accepted for publication in PLOS Neglected Tropical Diseases.

Best regards,

Gregory Deye

Associate Editor

Bruce Lee

Deputy Editor

---

## [Editor Report · Acceptance letter]

19 Nov 2021

Dear Dr. Kasparis,

We are delighted to inform you that your manuscript, "Improving Timeliness in the Neglected Tropical Diseases Preventive Chemotherapy Donation Supply Chain through Information Sharing: A Retrospective Empirical Analysis," has been formally accepted for publication in PLOS Neglected Tropical Diseases.

Best regards,

Shaden Kamhawi

co-Editor-in-Chief

Paul Brindley

co-Editor-in-Chief
